# Association of marital and parental status with stress, support, adherence, and quality of life among breast cancer women

Laraib Akram[1], Ghulam Abbas[2], Haris Khurram[3], Ayesha Aslam[4], Fawad Ahmad Randhawa[5], Abdullah A. Assiri[6], Shahid Shah[1]*

1 Faculty of Pharmaceutical Sciences, Department of Pharmacy Practice, Government College University Faisalabad, Faisalabad, Pakistan, 2 Faculty of Pharmaceutical Sciences, Department of Pharmaceutics, Government College University Faisalabad, Faisalabad, Pakistan, 3 FAST School of Management, National University of Computer and Emerging Science, Chiniot-Faisalabad Campus, Chiniot, Pakistan, 4 Department of Neurology, King Edward Medical University, Lahore, Pakistan, 5 Department of Endocrinology, Allama Iqbal Medical College Lahore, Lahore, Pakistan, 6 Department of Clinical Pharmacy, College of Pharmacy, King Khalid University, Abha, Saudi Arabia

* shahid.shah@gcuf.edu.pk

## Abstract

### Backgrounds

Breast cancer presents multifaceted challenges that extend beyond physical illness, profoundly influencing patients' psychological well-being, treatment adherence, and social relationships. Marital and parental status further affect these experiences by influencing coping capacity, emotional stability, and perceived support systems. This study aimed to assess how marital and parental status influence the lives of women with breast cancer, particularly their psychological well-being, support, treatment adherence, and overall quality of life (QoL).

### Methods

This cross-sectional study of 503 patients utilized validated research instruments for data collection: The Depression, Anxiety, and Stress Scale (DASS-21), the Multi-dimensional Scale of Perceived Social Support (MSPSS), the Medication Adherence Report Scale (MARS-5), and the WHOQOL-BREF. Descriptive statistics, Kruskal-Wallis tests, and Spearman's correlation (95% CI) were used as an exploratory data analysis. Multiple linear regression was utilized to measure the effect of different variables on the domains of QoL. The $p < 0.05$ was considered significant.

### Results

Across marital status, psychological distress, notably depression (median = 16.5, Q1-Q3 = 15–22.8, $p > 0.001$), was predominantly observed in divorced/widowed

**Data availability statement:** All relevant data are within the paper and its Supplementary file.

**Funding:** Deanship of Research and Graduate Studies at King Khalid University for funding this work through the Large Research Project under grant number RGP2/676/46 The funders had an active role not only in funding acquisition but also in providing resources and in the writing, review, and editing of the manuscript.

**Competing interests:** The authors have declared that no competing interests exist.

women, whereas single women exhibited better social support (median = 72, Q1-Q3 = 60–81.2, *p > 0.001*) and overall WHOQOL (median = 86, Q1-Q3 = 79–93, *p > 0.001*). Relating to parental status, mothers, particularly those with multiple children, experienced greater psychological distress (median = 16, Q1-Q3 = 12–18, *p > 0.001*). At the same time, childless women exhibited better social support (median = 72, Q1-Q3 = 60–76, *p > 0.001*) and QoL (median = 86, Q1-Q3 = 79–93, *p > 0.001*). A negative relationship of medication adherence with both social support and QoL was observed, while it showed a positive correlation with psychological distress in breast cancer patients, with a significant value (*p > 0.001*).

## Conclusions

In breast cancer patients, depression was highest among divorced/widowed women and mothers with multiple children, while single and childless women reported greater social support and QoL. Medication adherence showed a positive association with psychological distress and a negative correlation with both social support and QoL.

## Introduction

Breast cancer remains the most frequent cancer diagnosis among women and is a leading cause of cancer-related mortality worldwide [1]. According to the World Health Organization, 2.3 million new breast cancer cases are reported each year [2,3]. Recent evidence highlights a rising incidence among younger women aged 15–39 [4, 5], with studies from Pakistan reporting a marked rise compared to the Western population [6,7]. This transition underscores the necessity to assess how personal situations, like marital and parental status, greatly impact the lives of women with breast cancer [8,9]. These roles markedly impact crucial aspects of care, including stress, provision of social support, treatment compliance, and overall quality of life (QoL) [10,11]. Collectively, these interrelated factors emphasize the complicated nature of breast cancer management and the various challenges encountered by patients with breast cancer [12].

Despite considerable advancements in breast cancer treatment, the emotional, social, and behavioral aspects of the disease remain insufficiently addressed [13]. Women diagnosed with breast cancer have difficulties that extend beyond physical conditions, encompassing psychological distress, social support, medication adherence, and overall QoL [14–17]. In Pakistan, cultural norms and social expectations significantly influence how women navigate serious illnesses such as breast cancer [18]. Sociocultural factors affect caregiving roles and support, increasing psychological stress during treatment [19]. For example, family dynamics and gender roles have been shown to influence treatment decisions and caregiving responsibilities within the Pakistani context, reflecting broader social influences on health behavior [20]. These cultural norms

may affect perceived social support (PSS), engagement with treatment, and QoL, highlighting the importance of examining marital status and parental status within this setting [21, 22]. The parenting factors and relationships can either provide psychological strength through support from family or exacerbate stress induced by caregiving responsibilities [23].

Marital and parental statuses are recognized as important social determinants influencing women's experiences with breast cancer. Previous studies have generally examined these factors individually, focusing on outcomes such as stress, perceived support, treatment adherence, or QoL. However, limited attention has been given to evaluating the impact of both marital and parental status across all four of these outcomes within a single study [24]. Understanding these relationships is critical, as they can affect patients' psychological well-being, engagement with treatment, and overall QoL [25, 26]. The present study examines marital and parental statuses to provide a comprehensive perspective on their influence on psychological stress, PSS, treatment compliance, and QoL, highlighting the importance of considering social and familial factors in patient care.

## Materials and methods

### Study design and participants

This cross-sectional study was conducted from November 20th 2024, to 18th June 2025, in tertiary care hospitals of Punjab, Pakistan, and adhered to the Strengthening the Reporting of Observational Studies in Epidemiology (STROBE) guidelines for reporting observational studies. Data were collected from the oncology departments of 5 hospitals across Punjab, Pakistan, including 3 private and 2 government hospitals. The study initially recruited 613 breast cancer patients. 65 patients were eliminated due to incomplete or missing data, while an additional 45 patients failed to meet the inclusion and exclusion criteria. Moreover, the statistical power associated with this sample size exceeded 90%. Missing data were primarily due to participants' choice not to respond to certain items and submission of partially completed questionnaires at the time of data collection. Women aged 18–70 years with a confirmed diagnosis of breast cancer of any stage (I-IV) were eligible for inclusion. Participants were required to be receiving or have received standard breast cancer treatments (surgery, chemotherapy, radiotherapy, or a combination) and to have been diagnosed within a predefined period (1–2 years) to capture relevant treatment and psychosocial experiences. Both non-metastatic and metastatic cases were considered to ensure a representative sample. Participants with a documented history of severe psychiatric disorders or cognitive impairment that could interfere with the accurate completion of study instruments were excluded. Additionally, those with comorbid life-threatening conditions, such as advanced cardiac or renal failure, were excluded to avoid confounding effects on medication adherence, psychological stress, or treatment burden, ensuring that the data reflected breast cancer-specific outcomes. Recruitment was carried out through a non-probability sampling technique. After these exclusions, 503 patients were retained for the final analysis. Ethical approval was obtained from the Ethics Review Board of Government College University, Faisalabad, having ref. no. GCUF/ERC/497-A. Prior to participation, informed written consent was obtained from all participants. Participants were informed about the study objectives, procedures, and ethical considerations. They were assured that participation was voluntary and that they could withdraw at any time without any consequences. The study did not include minors; therefore, consent from parents or legal guardians was not required, and no waiver of informed consent was granted. Data were collected through structured, in-person interviews conducted by trained research assistants who were familiar with the study instruments and procedures. All interviews followed a standardized protocol to ensure consistency across participants. Questionnaires were administered by the interviewers in written form rather than self-administered, ensuring comprehension and completeness. The survey was conducted in the participants' native language, and validated translations of all instruments were used where necessary to maintain reliability and accuracy. The confidentiality and anonymity were strictly maintained throughout the study period.

## Instruments

### Depression, anxiety, and stress scale-21

The Depression, Anxiety, and Stress Scale-21 (DASS-21) was used to assess participants' psychosocial status [27]. It consists of three self-report subscales measuring depression, anxiety, and stress, each containing seven items. Higher scores indicate greater severity of symptoms in the respective domain [28].

### Multidimensional scale of perceived social support

The Multidimensional Scale of Perceived Social Support (MSPSS) was used to assess patients' perceived social support [29]. This 12-item questionnaire evaluates support from family, friends, and significant others [30]. Scores are calculated on a 7-point Likert scale, with 1–2.9 indicating low support, 3–5 indicating moderate support, and 5.1–7 indicating high support [31].

### Medication adherence report scale

The Medication Adherence Report Scale (MARS-5) is a five-question survey to assess a patient's medication adherence [32]. Five questions about forgetting, altering the dose, discontinuing, skipping, and reducing medication intake make up the MARS-5 questionnaire [33]. Higher self-reported adherence is shown by a greater MARS-5 score, with a range of 5−25. A cut-off score of less than 24 on the MARS-5 was suggested [34].

### The WHOQOL-BREF

The WHOQOL-BREF is a widely validated instrument for assessing QoL across four domains (physical, psychological, social, and environmental) and has demonstrated strong psychometric properties internationally [35]. A score less than 60 on the overall WHOQOL-BREF has been empirically supported as a meaningful threshold for poorer perceived QoL [36,37].

### Data analysis

Statistical analysis was performed using IBM SPSS Statistics version 27. Descriptive analysis, including frequency and percentages, was used to summarize the data. The Kruskal-Wallis test was used to compare the number of children-wise differences and marital status among the domains of psychological factors and WHOQOL. A correlation matrix with scatter plots was utilized to explore the relationship between the psychological and QoL variables. A multiple regression model with a forest plot was applied to measure the effect of different variables on the domains of QoL and to visualize the trend and patterns.

## Results

### Participants' demographics

A total of 503 patients completed the study. The majority of patients were between 25 and 34 years of age, constituting 35.98% of the sample, while 30.82% were aged 45 years or older, the second largest group. In terms of educational attainment, the majority of patients had no formal educational background, accounting for 44.53%. Based on marital status, 55.47% were married women and 40.16% were single. A small portion of patients (4.37%) were either divorced or widowed. Regarding parental status, approximately 57.66% of women were those with children, while the remaining 42.35% were childless. Most of the patients (87.28%) reported no family history of breast cancer, though some of them were unaware of their family medical background. Detailed demographics are illustrated in Table 1.

**Table 1. Sociodemographic characteristics of study participants with breast cancer.**

| Variable | Categories | No. % |
|---|---|---|
| **Age** | 18-24 | 35 (6.96) |
| | 25-34 | 181 (35.98) |
| | 35-44 | 132 (26.24) |
| | More than 45 | 155 (30.82) |
| **Highest level of education?** | Not at all | 224 (44.53) |
| | Primary School | 145 (28.83) |
| | Secondary School | 95 (18.89) |
| | Tertiary School | 39 (7.75) |
| **Marital status** | Divorced | 8 (1.59) |
| | Married | 279 (55.47) |
| | Single | 202 (40.16) |
| | Widow | 14 (2.78) |
| **Number of children** | 1 | 35 (6.96) |
| | 2 | 86 (17.1) |
| | 3 or more | 169 (33.6) |
| | None | 213 (42.35) |
| **Family history of cancer?** | No | 439 (87.28) |
| | Yes | 64 (12.72) |

### Marital status as a determinant of stress, social support, and QoL in breast cancer

Major differences among patients according to their marital status are presented in Table 2. Psychological stress, anxiety, and depression scores were the highest in divorced/widowed women, with depression (median = 16.5, Q1-Q3 = 15–22.8) being more prominent, followed by married women (median = 15, Q1-Q3 = 10.5–17), and the lowest in single women (median = 9.5, Q1-Q3 = 8–15), with a statistical significance of $p < 0001$. Higher social support scores were observed in single women (median = 72, Q1-Q3 = 60–81.2, $p < 0.001$). Across all QoL domains, single women scored the highest, with a WHOQOL score of (median = 86, Q1-Q3 = 79–93, $p < 0.001$); the only exception was the social relationships domain, where higher values were observed in married women (median = 12, Q1-Q3 = 10–12, $p < 0.001$).

#### Parental status as a determinant of stress, social support, and QoL in breast cancer

In Childless women, higher social support (median = 72, Q1-Q3 = 60–76) and WHOQOL scores (median = 86, Q1-Q3 = 79–93) were significantly observed ($P < 0.001$), reflecting better overall well-being. Conversely, mothers, particularly those with 3 or more children, exhibited heightened levels of stress, anxiety, and depression, with depression (median = 16, Q1-Q3 = 12–18) being the most significant ($p < 0.001$) of these psychological characteristics. The detailed interlinked variables and their respective outcomes are illustrated in Table 3.

### Interplay of stress, social support, treatment adherence, and QoL domains in women with breast cancer

Medication adherence displayed an interesting trend, showing a moderate negative correlation with social support (r = −0.359***) and all domains of WHOQOL, while a positive correlation with stress, anxiety, and depression, most notably a strong positive correlation with depression (r = 0.541***), indicating that, unlike social support, depression is closely influenced by adherence behavior. While observing social support, a negative correlation with psychosocial variables was observed, showing a moderate negative correlation with depression (r = −0.445***) and a strong positive correlation with WHOQOL (r = 0.639***), which indicated that perceiving support lowered psychosocial burden and enhanced QoL. The strongest positive correlation overall was observed between stress and depression (r = 0.928***), suggesting a tightly

**Table 2. Core health outcomes by marital status in breast cancer patients, median (Q1-Q3).**

| Variable | Marital Status | | | |
|---|---|---|---|---|
| | Single | Married | Divorced/Widow | P value |
| Medication adherence | 13 (10.2,16) | 14 (11.5,17) | 15.5 (12,18.8) | 0.17 |
| Social support | 72 (60,81.2) | 65 (60,72) | 49.5 (36,56.8) | < 0.001 |
| Stress | 11 (8.2,13) | 13 (9,15) | 16 (13,20.5) | < 0.001 |
| Anxiety | 4 (3,5) | 5 (4,6) | 6 (5,8) | < 0.001 |
| Depression | 9.5 (8,15) | 15 (10.5,17) | 16.5 (15,22.8) | < 0.001 |
| General health | 8 (8,8) | 7 (5,8) | 4 (2.2,5) | < 0.001 |
| Physical health | 24 (23,26) | 23 (20.5,26) | 18 (17.2,20) | < 0.001 |
| Psychological health | 17.8 (2.7) | 16.4 (2.6) | 13.1 (2.7) | < 0.001 |
| Social relationship | 11 (10,11) | 12 (10,12) | 6 (5,8.8) | < 0.001 |
| Environmental Conditions | 25.5 (21.2,29) | 25 (21,28) | 14 (12,21.8) | < 0.001 |
| WHOQOL | 86 (79,93) | 83 (72.5,90) | 56.5 (48.2,66.2) | < 0.001 |

**Table 3. Core health outcomes by parental status in breast cancer patients, median (Q1-Q3).**

| Variable | Child | | | |
|---|---|---|---|---|
| | Childless women | Women with children | | |
| | No | <3 | >=3 | P value |
| Medication adherence | 13 (11,16) | 14 (11,17) | 14 (12,17) | 0.263 |
| Social support | 72 (60,76) | 60 (57,72) | 64 (60,72) | < 0.001 |
| Stress | 11 (9,13) | 13 (9,15) | 14 (10,15) | < 0.001 |
| Anxiety | 4 (3,5) | 5 (4,6) | 5 (4,6) | < 0.001 |
| Depression | 10 (8,15) | 15 (10,17) | 16 (12,18) | < 0.001 |
| General health | 8 (7,8) | 7 (5,8) | 7 (5,8) | < 0.001 |
| Physical health | 24 (22,26) | 23 (20,26) | 23 (20,25) | 0.002 |
| Psychological health | 17.7 (2.7) | 16.4 (2.9) | 16 (2.6) | < 0.001 |
| Social relationship | 11 (10,11) | 12 (9,12) | 11 (9,12) | 0.071 |
| Environmental Conditions | 26 (21,29) | 26 (20,28) | 25 (21,27) | 0.023 |
| WHOQOL | 86 (79,93) | 83 (70,90) | 81 (69,90) | < 0.001 |

intertwined relationship, where rising stress levels markedly increase depressive symptoms among patients. A comprehensive overview of all correlation patterns is visually mapped out in Fig 1.

### Factors predicting WHOQOL domain scores in breast cancer

Multivariable regression analyses were conducted for each QoL domain (Environment, Physical Health, Psychological Health, Social Relationship, and WHOQOL), including child status, family history, marital status, medication adherence, social support, stress, anxiety, and depression as predictors. Variables were selected based on prior evidence of their impact on psychosocial outcomes in breast cancer patients [38,39]. Forest plots (Fig 2) show the estimated effects and 95% confidence intervals for each predictor.

In the environmental domain (Fig 2A), social support exhibited a positive correlation ($p < 0.001$), corresponding to the most significant overall effect on WHOQOL ($p < 0.001$), indicating its cumulative impact across all QoL domains. This indicates that increased social support leads to better overall QoL and improves how patients feel about their surroundings.

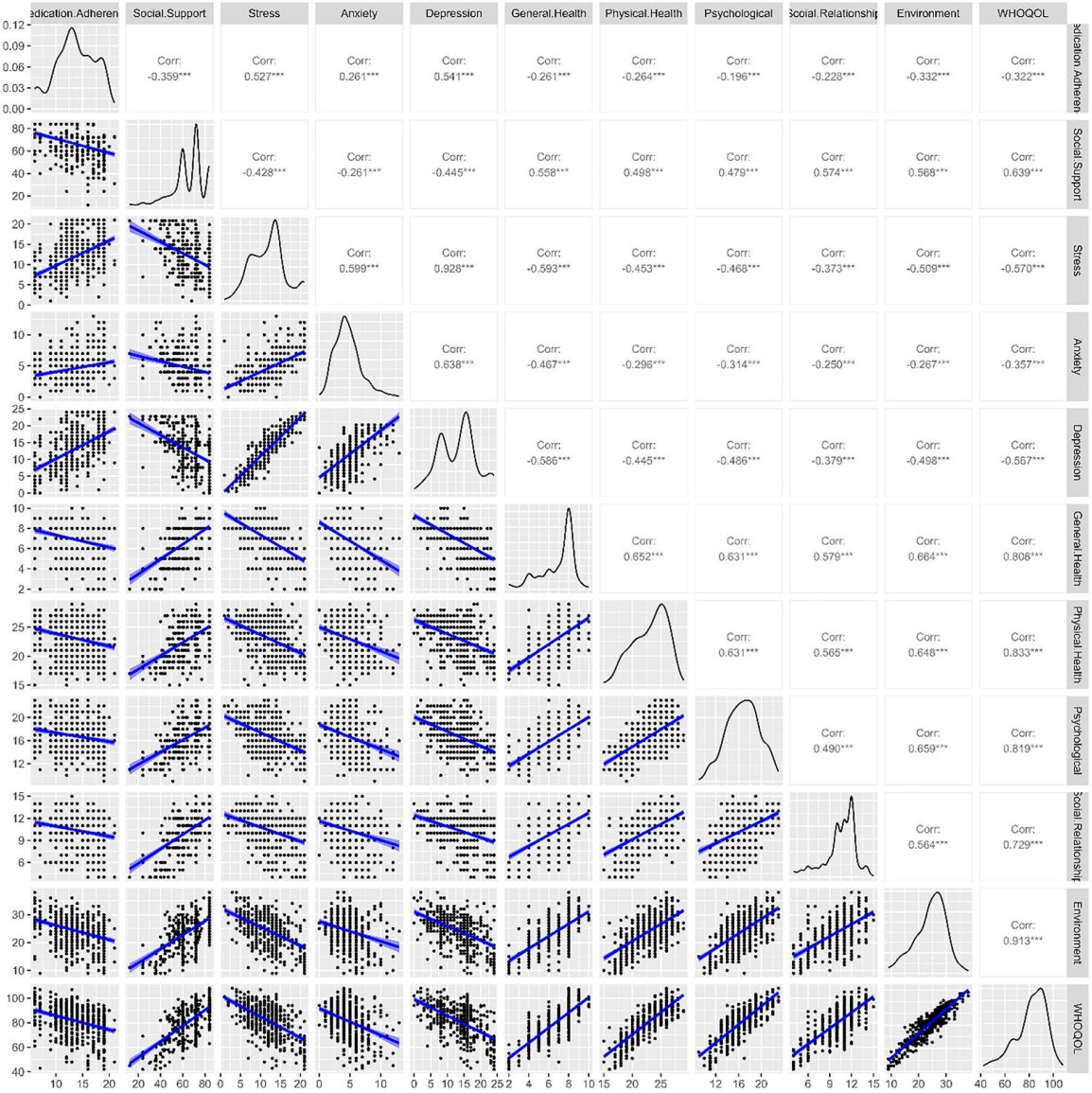

**Fig 1. Relationship between medication adherence, stress, social support, and QoL in breast cancer.** Note: ***p < 0.001.

Stress was reported to have a lower but significant negative correlation in both environmental (*p = 0.007*) and physical health domains (*p = 0.031*), as represented in Fig 2B, indicating that higher stress levels are linked to poorer physical well-being and a less favorable perception of the surroundings.

Depression was the most prominent predictor, with a notable negative association in the psychological health domain (*p = 0.004*), as observed in Fig 2C. This indicates that higher depression levels greatly reduce emotional stability, self-esteem, and the ability to cope with daily challenges. In marital status, divorced/widowed women notably presented lower but negative scores in the physical health domain (*p = 0.006*), social relationship domain (*p = 0.001*), observed in Fig 2D, with notable negative value in the psychological domain (*p = 0.003*) and overall WHOQOL (*p < 0.001*), as shown in Fig 2E. This suggests that divorced/widowed patients exhibited overall lower QoL.

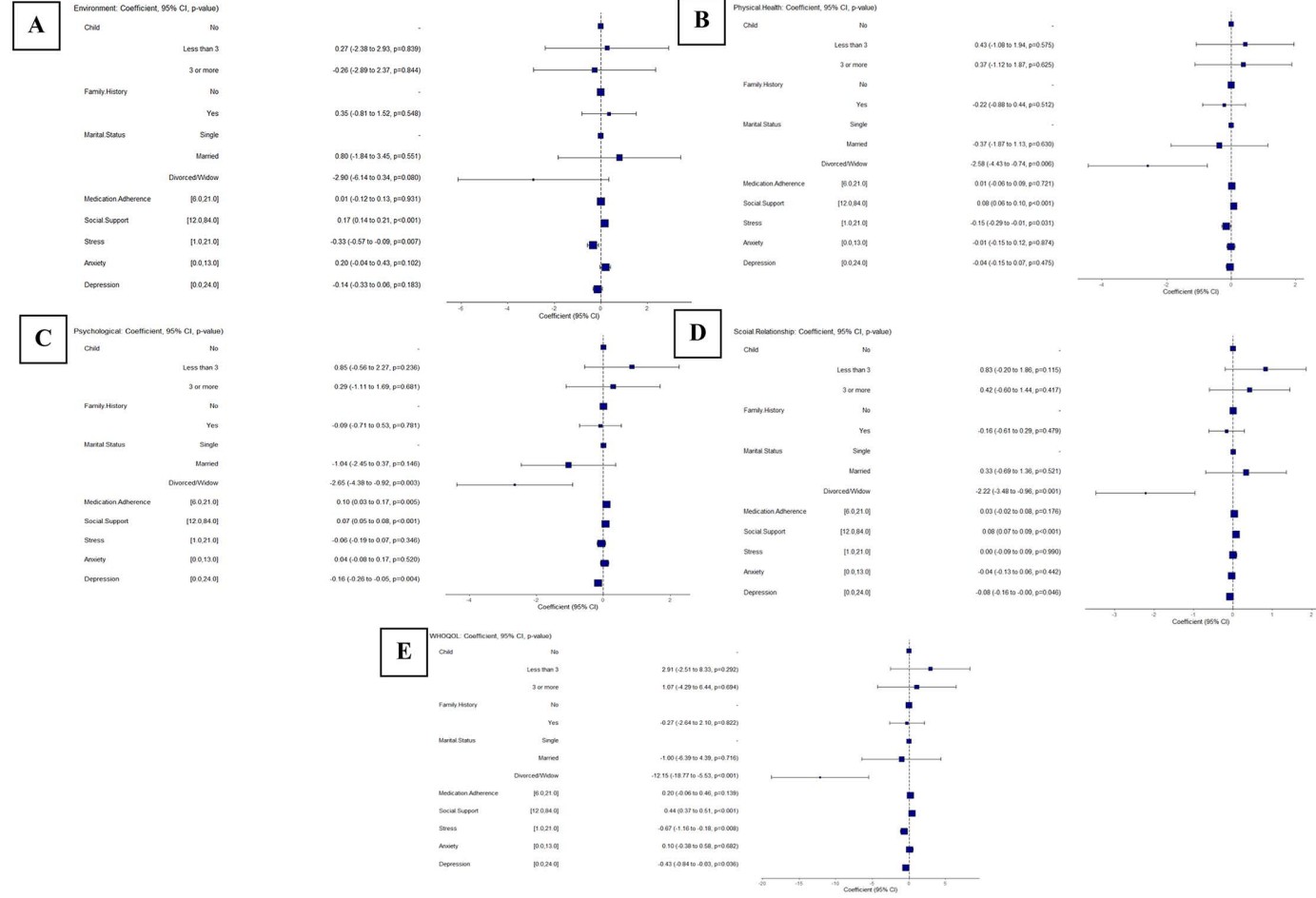

**Fig 2. Key determinants of QoL domains (A = Environmental, B = Physical, C = Psychological, D = Social Relationships, E = Overall WHOQOL.**

## Discussion

In this cross-sectional study, regarding marital status, we observed that single women reported higher levels of social support and better QoL. These findings are consistent with a Palestinian study, where single women similarly reported greater PSS and better QoL, likely due to ongoing emotional and practical support from family and friends [40]. This highlights that stronger social networks may play a key role in enhancing psychosocial outcomes in women with breast cancer [41]. The highest levels of depression were observed in divorced/widowed patients. These findings align with a study from China [42], where divorced patients also reported greater depressive symptoms. These results suggest that divorced/widowed patients may be particularly vulnerable to depression, potentially due to lower emotional and social support compared with other marital groups [43].

Regarding parental status, childless women experienced better psychological well-being compared to mothers, a pattern associated with greater social support and better QoL. This aligns with findings from Poland, where childless women reported higher satisfaction and improved QoL, likely due to reduced caregiving responsibilities [44]. This suggests that reduced caregiving responsibilities may contribute to better psychosocial outcomes in childless women [45]. In contrary, married women, particularly those with 3 or more children, had worse levels of depression. Consistent with our findings, a

Moroccan study showed that mothers with multiple children faced increased emotional discomfort, exacerbating psychological burden during breast cancer [46]. These findings suggest that managing parenting responsibilities alongside illness issues may elevate psychological stress [21].

Social support was consistently a positive predictor across all domains of QoL, while depression was negatively associated with each domain, particularly psychological health, highlighting its detrimental impact on mental and emotional well-being. Previous studies have similarly reported that long-term depressive symptoms worsen QoL, especially in patients undergoing chemotherapy [47,48]. Overall, these results show that strong social support improves QoL, whereas elevated depression and insufficient support markedly reduce it [49].

Interestingly, medication adherence was negatively associated with social support and all QoL domains. This is consistent with prior research showing that social support does not always improve adherence and may, in some chronic conditions, have complex or counterintuitive effects [50]. Similar findings from Australia indicate that higher treatment demands and complex regimens can negatively impact the relationship between medication adherence and QoL [51]. This indicates that higher social support or complex treatment demands may sometimes lower medication adherence and negatively affect QoL. It is possibly due to greater dependence on others or treatment complexity can increase patient burden and stress [52].

This indicates that higher social support or more complex treatment demands may sometimes reduce medication adherence and negatively affect quality of life, possibly because greater dependence on others or treatment complexity can increase patient burden and stress.

In our study, medication adherence was positively correlated with stress, anxiety, and depression, particularly depression, aligning with a South Korean study, where higher depression was paradoxically linked to greater medication adherence in breast cancer patients [12]. This suggests that psychological distress, such as anxiety or fear of disease progression, may motivate adherence in life-threatening conditions like cancer. This is possibly because patients with higher symptom burden or greater perceived risk are more likely to follow prescribed treatments, which can impact their QoL [53].

Our findings provide insight into how psychosocial distress manifests differently across marital and parental statuses in women with breast cancer. The study underscores the importance of assessing emotional distress and social support to develop individualized interventions that may improve treatment adherence and enhance QoL outcomes.

## Limitations

The cross-sectional design limits causal conclusions and long-term assessment but provides valuable insight into the relationships between marital and parental status, psychological distress, social support, medication adherence, and QoL. The use of non-probability sampling from hospitals in Punjab, Pakistan, may limit generalizability to other populations. In addition, unmeasured confounders such as socioeconomic status, education, income, employment, and access to healthcare may have influenced the findings. Future research should use longitudinal and cross-cultural designs with diverse samples to better understand how marital and parental status, social support, psychological distress, and medication adherence affect overall QoL.

## Conclusion

The psychosocial differences among women with breast cancer revealed that single and childless women reported better QoL and social support, whereas mothers and divorced/widowed participants experienced increased psychological discomfort. A higher level of psychological distress resulted in reduced adherence rates, while elevated medication adherence was observed in patients with lower PSS and QoL. This interplay between psychological factors, medication adherence, and PSS highlights the multifaceted nature of the breast cancer experience. Recognizing these dynamics is essential for understanding how marital and parental status influence emotional well-being and treatment engagement.

Recognizing the influence of marital and parental status on these outcomes can guide clinicians in developing more individualized and effective psychosocial care strategies for women with breast cancer.

## Supporting information

**S1 Dataset. Data.**
(XLSX)

## Author contributions

**Conceptualization:** Ghulam Abbas, Shahid Shah.

**Data curation:** Laraib Akram, Ayesha Aslam.

**Formal analysis:** Haris Khurram, Ayesha Aslam.

**Funding acquisition:** Abdullah A Assiri.

**Investigation:** Ghulam Abbas, Shahid Shah.

**Methodology:** Laraib Akram, Ayesha Aslam.

**Resources:** Ghulam Abbas, Abdullah A Assiri.

**Supervision:** Haris Khurram, Shahid Shah.

**Validation:** Ghulam Abbas, Haris Khurram.

**Visualization:** Haris Khurram, Fawad Ahmad Randhawa.

**Writing – original draft:** Laraib Akram.

**Writing – review & editing:** Fawad Ahmad Randhawa, Abdullah A Assiri, Shahid Shah.

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
