## [Decision Letter · Decision Letter 0]

11 Dec 2025

Thank you for submitting your manuscript to PLOS ONE. After careful consideration, we feel that it has merit but does not fully meet PLOS ONE’s publication criteria as it currently stands. Therefore, we invite you to submit a revised version of the manuscript that addresses the points raised during the review process.

plosone@plos.org . . A rebuttal letter that responds to each point raised by the academic editor and reviewer(s). You should upload this letter as a separate file labeled 'Response to Reviewers'.A marked-up copy of your manuscript that highlights changes made to the original version. You should upload this as a separate file labeled 'Revised Manuscript with Track Changes'.An unmarked version of your revised paper without tracked changes. You should upload this as a separate file labeled 'Manuscript'.

We look forward to receiving your revised manuscript.

Kind regards,

Mukhtiar Baig, Ph.D.

Academic Editor

PLOS One

Journal Requirements:

[Deanship of Research and Graduate Studies at King Khalid University for funding this work through the Large Research Project under grant number RGP2/676/46].

[The authors extend their appreciation to the Deanship of Research and Graduate Studies at King Khalid University for funding this work through the Large Research Project under grant number RGP2/676/46]

[Deanship of Research and Graduate Studies at King Khalid University for funding this work through the Large Research Project under grant number RGP2/676/46]

[NO authors have competing interests].

7. Please include a separate caption for each figure in your manuscript.

8. Please include captions for your Supporting Information files at the end of your manuscript, and update any in-text citations to match accordingly. Please see our Supporting Information guidelines for more information: http://journals.plos.org/plosone/s/supporting-information ..

Reviewers' comments:

Reviewer's Responses to Questions

**Comments to the Author**

1. Is the manuscript technically sound, and do the data support the conclusions?

Reviewer #1: Yes

Reviewer #2: Partly

Reviewer #3: Yes

2. Has the statistical analysis been performed appropriately and rigorously?

Reviewer #1: Yes

Reviewer #2: Yes

Reviewer #3: Yes

3. Have the authors made all data underlying the findings in their manuscript fully available?

Reviewer #1: Yes

Reviewer #2: Yes

Reviewer #3: Yes

4. Is the manuscript presented in an intelligible fashion and written in standard English?

Reviewer #1: Yes

Reviewer #2: Yes

Reviewer #3: Yes

Reviewer #1: 1. Is the manuscript technically sound, and do the data support the conclusions?

The study is well-designed and methodologically solid, but some aspects need further clarification or improvement. The research question is relevant and addresses essential gaps in the psychosocial aspects of breast cancer care. The focus on marital and parental status as factors influencing QoL and psychological outcomes is innovative and provides a valuable perspective to existing literature. The sample size is adequate for a cross-sectional study and should allow general conclusions within the study's context. The use of validated instruments is a strength, as it guarantees reliable measurement of the variables.

The manuscript lacks a detailed explanation of how the sample size was determined. Given that this is a cross-sectional study, it would be helpful to include a justification for the sample size, based on power analysis or similar statistical reasoning.

The study doesn’t sufficiently discuss potential confounders, such as socioeconomic status, education, or other personal factors that could influence the results. For example, factors such as income, employment, or healthcare access may also play a significant role in the observed outcomes. If this is not addressed, it must be acknowledged as a limitation of the work. The conclusions are well-supported by the data, but the discussion needs to integrate these findings with existing literature on marital and parental status in breast cancer patients.

Also, many figures were not added and this should be addressed and added.

2. Has the statistical analysis been performed appropriately and rigorously?

Yes, the statistical analysis appears rigorous, but a few points need to be addressed. The regression analysis should be better described. Authors need to explain which variables were included in the regression model and the rationale behind the selection.

3. Have the authors made all data underlying the findings in their manuscript fully available?

The manuscript includes data availability statements that align with PLOS ONE’s data-sharing policies.

4. Is the manuscript presented in an intelligible fashion and written in standard English?

The manuscript is well-organized, but several areas could be clarified and improved in grammar and flow. It has a logical structure, making it easy to follow the progression between sections.

Some sentences need more clarity. For example, in the "Results" section, phrases like "psychological discomfort was most pronounced in mothers" could be clearer. Consider using “mothers, particularly those with multiple children, experienced greater psychological distress.” There are some minor grammatical errors and awkward phrasing that should be proofread and fixed.

1. Is the manuscript technically sound, and do the data support the conclusions?

The study is well-designed and methodologically solid, but some aspects need further clarification or improvement. The research question is relevant and addresses essential gaps in the psychosocial aspects of breast cancer care. The focus on marital and parental status as factors influencing QoL and psychological outcomes is innovative and provides a valuable perspective to existing literature. The sample size is adequate for a cross-sectional study and should allow general conclusions within the study's context. The use of validated instruments is a strength, as it guarantees reliable measurement of the variables.

The manuscript lacks a detailed explanation of how the sample size was determined. Given that this is a cross-sectional study, it would be helpful to include a justification for the sample size, based on power analysis or similar statistical reasoning.

The study doesn’t sufficiently discuss potential confounders, such as socioeconomic status, education, or other personal factors that could influence the results. For example, factors such as income, employment, or healthcare access may also play a significant role in the observed outcomes. If this is not addressed, it must be acknowledged as a limitation of the work. The conclusions are well-supported by the data, but the discussion needs to integrate these findings with existing literature on marital and parental status in breast cancer patients.

Also, many figures were not added and this should be addressed and added.

2. Has the statistical analysis been performed appropriately and rigorously?

Yes, the statistical analysis appears rigorous, but a few points need to be addressed. The regression analysis should be better described. It would be helpful to explain which variables were included in the regression model and the rationale behind the selection.

3. Have the authors made all data underlying the findings in their manuscript fully available?

The manuscript includes data availability statements that align with PLOS ONE’s data-sharing policies.

4. Is the manuscript presented in an intelligible fashion and written in standard English?

The manuscript is well-organized, but several areas could be clarified and improved in grammar and flow. It has a logical structure, making it easy to follow the progression between sections. Generally, the writing is clear, and the abstract effectively summarizes the study.

Some sentences need more clarity. For example, in the "Results" section, phrases like "psychological discomfort was most pronounced in mothers" could be clearer. Consider using “mothers, particularly those with multiple children, experienced greater psychological distress.” There are some minor grammatical errors and awkward phrasing that should be proofread and fixed.

LIMITATIONS THAT NEED TO BE INCLUDED

The study's cross-sectional design captures data at a single point in time, limiting the ability to draw causal conclusions or infer long-term trends. Longitudinal studies provide a better understanding of how marital and parental status influence psychological distress, social support, medication adherence, and QoL over time. The study used non-probability sampling from five hospitals in Punjab, Pakistan, possibly causing selection bias and limiting generalizability. Results may not apply to women in rural areas, different cultures, or without healthcare access. The study didn't fully control for confounders like socioeconomic status, education, comorbidities, or treatment type, which could influence psychological outcomes and quality of life. Future research should include these variables to better isolate effects of marital and parental status.

The study's setting in Punjab, Pakistan, may restrict the external validity of the findings. Cultural norms and expectations concerning marriage, family, and caregiving differ significantly across regions. Consequently, the results may not be generalizable to populations in different cultural or geographic contexts. Conducting cross-cultural comparisons would be beneficial in examining how these factors influence women with breast cancer in various settings. Although the study included 503 patients in the final analysis, 110 patients were excluded due to incomplete or missing data. The reasons for missing data are not specified, and the potential impact on the results remains unclear. It should be acknowledged as a limitation.

Reviewer #2: Greetings, thank you for giving me the opportunity to review this manuscript, I have some concerns that I will explain below.

Firstly, in the methods part, exclusion criteria, the authors state the patients with co-morbid life threatening conditions were excluded. What does this sentence mean? An example on such condition should be provided, moreover, an explanation to why authors made this decision, and how is it going to affect the data generated should also be provided in the main text.

Secondly, according to the data released by the World Health Organization, International Agency for Research on Cancer on 24th of FEBRUARY, 2025. ( Globally, most breast cancer cases and deaths occur in individuals aged 50 years and older, who account for 71% of new cases and 79% of deaths ). In this study, the sample included a majority of 181 patients out of 503 that were aged between 25-34, compared to 155 patients aged greater that 45. These numbers contradict the known data regarding the common age of breast cancer, I was hoping to find a part that addresses this issue, even in the discussion section but the authors seemed ignorant of the idea, it was not mentioned as a limitation as well. Such numbers of cases in a young age are strongly linked to family history or genetic mutations like BRCA 1 gene mutation. However, the majority of the responses reported denied having family history of cancer.

This issue is very crucial, and if not properly addressed it may result in an unrepresentative sample and thus a biased study. I kindly ask the authors to investigate why larger numbers of cases in a younger ( unexpected age ) were involved in their study sample. Were all participants from the same geographical area? Are the hospitals included in the study known for treating younger aged patients? What makes these similar\ different from the global population?

Thirdly, line 210-213, if this conclusion is mentioned by other studies in the existing literature, please reference them. Same goes for lines 215-218, 224-225, 229-230.

Fourthly, line 232, it seems like there is a missing part, depression negatively associated with what?

Lines 248-249 contradict what is mentioned in the Indonesian study mentioned in line 247.

Thank you again for your submission, I hope the authors take these points into consideration.

Reviewer #3: The topic is important, but the research needs some critical corrections.

**Do you want your identity to be public for this peer review?** For information about this choice, including consent withdrawal, please see our For information about this choice, including consent withdrawal, please see our Privacy Policy .

Reviewer #1: No

Reviewer #2: No

Reviewer #3: No

---

## [Author Response · Author response to Decision Letter 1]

5 Jan 2026

We sincerely thank the Editor and the reviewers for their valuable comments and suggestions on our manuscript. We have carefully considered all comments and revised the manuscript ac-cordingly.

Journal Requirements

Query 1. Please ensure that your manuscript meets PLOS ONE's style requirements, including those for file naming. The PLOS ONE style templates can be found at

Response: All the necessary changes have been made in accordance with the PLOS ONE style requirements, including those related to file naming

Query 2. Please provide additional details regarding participant consent. In the ethics statement in the Methods and online submission information, please ensure that you have specified (1) whether consent was informed and (2) what type you obtained (for instance, written or verbal, and if verbal, how it was documented and witnessed). If your study included minors, state whether you obtained consent from parents or guardians. If the need for consent was waived by the ethics committee, please include this information.

Response: Page # 6, Line # 129-131

Details regarding consent form has been updated in manuscript.

Query 3. Thank you for stating the following financial disclosure:

[Deanship of Research and Graduate Studies at King Khalid University for funding this work through the Large Research Project under grant number RGP2/676/46].

Response: The funder played a role in the manuscript writing, review, and editing, as already stated in the Author Contributions section

Query 4. Thank you for stating the following in the Acknowledgments Section of your manu-script:

[The authors extend their appreciation to the Deanship of Research and Graduate Studies at King Khalid University for funding this work through the Large Research Project under grant number RGP2/676/46]

We note that you have provided funding information that is currently declared in your Funding Statement. However, funding information should not appear in the Acknowledgments section or other areas of your manuscript. We will only publish funding information present in the Fund-ing Statement section of the online submission form.

[Deanship of Research and Graduate Studies at King Khalid University for funding this work through the Large Research Project under grant number RGP2/676/46]

Response: We have removed funding details from the acknowledgment section as per the re-quirement.

Query 5. Thank you for stating the following in your Competing Interests section:

[NO authors have competing interests].

Please complete your Competing Interests on the online submission form to state any Compet-ing Interests. If you have no competing interests, please state "The authors have declared that no competing interests exist.", as detailed online in our guide for authors at http://journals.plos.org/plosone/s/submit-now

Response: Competing interest section has been updated as per the guidelines.

Query 6. When completing the data availability statement of the submission form, you indicat-ed that you will make your data available on acceptance. We strongly recommend all authors decide on a data sharing plan before acceptance, as the process can be lengthy and hold up pub-lication timelines. Please note that, though access restrictions are acceptable now, your entire data will need to be made freely accessible if your manuscript is accepted for publication. This policy applies to all data except where public deposition would breach compliance with the pro-tocol approved by your research ethics board. If you are unable to adhere to our open data poli-cy, please kindly revise your statement to explain your reasoning and we will seek the editor's input on an exemption. Please be assured that, once you have provided your new statement, the assessment of your exemption will not hold up the peer review process.

Response: The raw data have been shared in accordance with PLOS ONE guidelines

Query 7. Please include a separate caption for each figure in your manuscript.

Response: Separate captions for each figure have been included in the manuscript as per the instructions pages 12–13; lines 211 and 234.

Captions of the figures have been included in the manuscript as per the requirements.

Query 8. Please include captions for your Supporting Information files at the end of your manu-script, and update any in-text citations to match accordingly. Please see our Supporting Infor-mation guidelines for more information: http://journals.plos.org/plosone/s/supporting-information.

Response: No supporting information files are included; therefore, no captions or in-text cita-tions are required.

Query 9: If the reviewer comments include a recommendation to cite specific previously pub-lished works, please review and evaluate these publications to determine whether they are rele-vant and should be cited. There is no requirement to cite these works unless the editor has indi-cated otherwise.

Response: The reviewer comments did not include any recommendations to cite specific previ-ously published works.

Reviewer # 1

Query # 1. Is the manuscript technically sound, and do the data support the conclusions?

The study is well-designed and methodologically solid, but some aspects need further clarifica-tion or improvement. The research question is relevant and addresses essential gaps in the psy-chosocial aspects of breast cancer care. The focus on marital and parental status as factors in-fluencing QoL and psychological outcomes is innovative and provides a valuable perspective to existing literature. The sample size is adequate for a cross-sectional study and should allow general conclusions within the study's context. The use of validated instruments is a strength, as it guarantees reliable measurement of the variables.

The manuscript lacks a detailed explanation of how the sample size was determined. Given that this is a cross-sectional study, it would be helpful to include a justification for the sample size, based on power analysis or similar statistical reasoning.

The study doesn’t sufficiently discuss potential confounders, such as socioeconomic status, edu-cation, or other personal factors that could influence the results. For example, factors such as income, employment, or healthcare access may also play a significant role in the observed out-comes. If this is not addressed, it must be acknowledged as a limitation of the work. The con-clusions are well-supported by the data, but the discussion needs to integrate these findings with existing literature on marital and parental status in breast cancer patients.

Also, many figures were not added and this should be addressed and added.

Response: This cross-sectional study included all eligible women with breast cancer who visit-ed the study hospital during the data collection period. A total of 503 participants were included in the analysis, providing statistical power greater than 90% to detect meaningful associations as stated in manuscript page # 5, line # 113-114.

We acknowledge that unmeasured confounders, including socioeconomic status, education, in-come, employment, and access to healthcare, may have influenced the findings. This limitation has now been added to the manuscript, and future studies are recommended to address these factors for a more comprehensive understanding.

The discussion has been updated to include relevant literature on marital and parental status in breast cancer patients as stated in manuscript page # 13-15 line # 236-277.

Query # 2. Has the statistical analysis been performed appropriately and rigorously?

Yes, the statistical analysis appears rigorous, but a few points need to be addressed. The regres-sion analysis should be better described. Authors need to explain which variables were included in the regression model and the rationale behind the selection.

Response: The regression analysis has been updated to include a detailed description of the var-iables in the model and the rationale for their selection as stated in manuscript page # 12, line # 213-218.

Query # 3. Have the authors made all data underlying the findings in their manuscript ful-ly available?

The manuscript includes data availability statements that align with PLOS ONE’s data-sharing policies.

Query # 4. Is the manuscript presented in an intelligible fashion and written in standard English?

The manuscript is well-organized, but several areas could be clarified and improved in grammar and flow. It has a logical structure, making it easy to follow the progression between sections.

Some sentences need more clarity. For example, in the "Results" section, phrases like "psycho-logical discomfort was most pronounced in mothers" could be clearer. Consider using “mothers, particularly those with multiple children, experienced greater psychological distress.” There are some minor grammatical errors and awkward phrasing that should be proofread and fixed.

Response: The result section has been updated as per the suggestions as stated in manuscript page # 2, line# 45-46.

The manuscript has been carefully proofread, and the minor grammatical errors and awkward phrasing have been corrected following the page # 4, 5, 6-7, 12, 13, 14, 16, line # 76, 107,129, 140-141, 219, 223-224, 227, 231-232, 239-241, 256-258, 260, 292.

Query # 5. Limitations That Need to Be Included

The study's cross-sectional design captures data at a single point in time, limiting the ability to draw causal conclusions or infer long-term trends. Longitudinal studies provide a better under-standing of how marital and parental status influence psychological distress, social support, medication adherence, and QoL over time. The study used non-probability sampling from five hospitals in Punjab, Pakistan, possibly causing selection bias and limiting generalizability. Re-sults may not apply to women in rural areas, different cultures, or without healthcare access. The study didn't fully control for confounders like socioeconomic status, education, comorbidi-ties, or treatment type, which could influence psychological outcomes and quality of life. Future research should include these variables to better isolate effects of marital and parental status.

The study's setting in Punjab, Pakistan, may restrict the external validity of the findings. Cultur-al norms and expectations concerning marriage, family, and caregiving differ significantly across regions. Consequently, the results may not be generalizable to populations in different cultural or geographic contexts. Conducting cross-cultural comparisons would be beneficial in examining how these factors influence women with breast cancer in various settings. Although the study included 503 patients in the final analysis, 110 patients were excluded due to incom-plete or missing data. The reasons for missing data are not specified, and the potential impact on the results remains unclear. It should be acknowledged as a limitation.

Response: The limitations section has been updated in accordance with the reviewer’s sugges-tions as stated in manuscript page # 15, line # 279-285.

The handling of missing data has been clarified in the Materials and Methods section, where the exclusion of 110 patients due to incomplete or missing data is now specified following page # 5, line# 114-116.

Reviewer # 2

Query # 1. In the methods part, exclusion criteria, the authors state the patients with co-morbid life-threatening conditions were excluded. What does this sentence mean? An example on such condition should be provided, moreover, an explanation to why authors made this decision, and how is it going to affect the data generated should also be provided in the main text.

Response: The patients with comorbid life-threatening conditions, such as advanced cardiac or renal failure, were excluded to avoid confounding effects on medication adherence, psychologi-cal stress, or treatment burden, ensuring that the data reflected breast cancer-specific outcomes as stated in manuscript page # 6, line# 123-126.

Query # 2. According to the data released by the World Health Organization, International Agency for Research on Cancer on 24th of FEBRUARY, 2025. ( Globally, most breast cancer cases and deaths occur in individuals aged 50 years and older, who account for 71% of new cas-es and 79% of deaths ). In this study, the sample included a majority of 181 patients out of 503 that were aged between 25-34, compared to 155 patients aged greater that 45. These numbers contradict the known data regarding the common age of breast cancer, I was hoping to find a part that addresses this issue, even in the discussion section but the authors seemed ignorant of the idea, it was not mentioned as a limitation as well. Such numbers of cases in a young age are strongly linked to family history or genetic mutations like BRCA 1 gene mutation. However, the majority of the responses reported denied having family history of cancer.

This issue is very crucial, and if not properly addressed it may result in an unrepresentative sample and thus a biased study. I kindly ask the authors to investigate why larger numbers of cases in a younger ( unexpected age ) were involved in their study sample. Were all participants from the same geographical area? Are the hospitals included in the study known for treating younger aged patients? What makes these similar\ different from the global population?

Response: We appreciate the insightful comment regarding the age distribution in our sample. While global data from the World Health Organization’s International Agency for Research on Cancer indicate that most breast cancer cases and deaths occur in individuals aged 50 years and older, considerable regional variation in breast cancer age distribution exists globally, with dis-tinct patterns evident across Asian populations. Recent studies from the Asia-specific region indicate that a substantial proportion of breast cancer patients present at younger ages compared with Western cohorts, with peak incidence often occurring in the fourth decade of life rather than later decades(1-3).

Although early-onset breast cancer has been associated with family history and genetic muta-tions such as BRCA1/2, many younger patients do not report a known genetic predisposition. This may partly reflect limited awareness of family medical history and restricted access to ge-netic testing and medical records in the study setting(4). All participants were recruited from the same geographical region and referral hospitals, which may receive a higher proportion of younger patients with more aggressive disease, potentially leading to overrepresentation of ear-ly-onset cases. Lower screening uptake and awareness may also contribute to under-reporting of family history. Notably, the World Health Organization (2024) reports that breast cancer com-monly occurs in women without identifiable risk factors such as family history or known genet-ic mutations(5, 6).

References:

1. Soomro R, Faridi S, Khurshaidi N, Zahid N, Mamshad I. Age and stage of breast cancer in Pakistan: An experience at a tertiary care center. J Pak Med Assoc. 2018;68(11):1682-5.

2. Zaheer S, Shah N, Maqbool SA, Soomro NM. Estimates of past and future time trends in age-specific

---

## [Decision Letter · Decision Letter 1]

15 Feb 2026

Dear Dr. Shah,

We look forward to receiving your revised manuscript.

Kind regards,

Mukhtiar Baig, Ph.D.

Academic Editor

PLOS One

Journal Requirements:

Reviewers' comments:

Reviewer's Responses to Questions

**Comments to the Author**

Reviewer #1: (No Response)

Reviewer #2: All comments have been addressed

2. Is the manuscript technically sound, and do the data support the conclusions?

Reviewer #1: Yes

Reviewer #2: Yes

3. Has the statistical analysis been performed appropriately and rigorously?

Reviewer #1: Yes

Reviewer #2: Yes

4. Have the authors made all data underlying the findings in their manuscript fully available?

Reviewer #1: Yes

Reviewer #2: No

5. Is the manuscript presented in an intelligible fashion and written in standard English?

Reviewer #1: Yes

Reviewer #2: Yes

Reviewer #1: Minor comments

The sentences in Lines 87-90 should be cited as well as sentences in line 99-102. Line 348 states 'Future longitudinal and cross-cultural studies may further clarify these relationships'. This statement is ambiguous and so there should be a mini section that addresses the future research directions in the area of this research.

Reviewer #2: Thank you for addressing the issues and doing modifications accordingly.

My major concern was the generalizability of the results of a tertiary hospital data, where cases can differ from global estimates and more aggressive disease form are present. Since the authors declared that the data do not provide generalizable data at the moment and more studies are needed, then the main issue has been addressed and paper is ready for publication.

**Do you want your identity to be public for this peer review?** For information about this choice, including consent withdrawal, please see our For information about this choice, including consent withdrawal, please see our Privacy Policy .

Reviewer #1: No

Reviewer #2: No

---

## [Author Response · Author response to Decision Letter 2]

4 Mar 2026

We want to express our thanks to the Academic Editor for allowing us to revise and improve our manuscript. We truly appreciate the time and consideration you and the reviewers have invested in evaluating our work. The feedback provided is valuable, and we are committed to address all comments carefully to strengthen the quality and clarity of the manuscript. We are grateful for the chance to refine the study in line with the journal’s standards.

Journal Requirements

Query # 1. If the reviewer comments include a recommendation to cite specific previously published works, please review and evaluate these publications to determine whether they are relevant and should be cited. There is no requirement to cite these works unless the editor has indicated otherwise.

Response: No specific previously published works were recommended by the reviewers for citation.

Query # 2. Please review your reference list to ensure that it is complete and correct. If you have cited papers that have been retracted, please include the rationale for doing so in the manuscript text, or remove these references and replace them with relevant current references. Any changes to the reference list should be mentioned in the rebuttal letter that accompanies your revised manuscript. If you need to cite a retracted article, indicate the article’s retracted status in the References list and also include a citation and full reference for the retraction notice.

Response: The manuscript does not contain any retracted references, so no action regarding such articles is required. The changes made to the reference list have been duly noted and mentioned in the rebuttal letter as per the recommendations.

Note: We would like to inform you that the affiliation for one of our authors, Harris Khurram, has been updated. Kindly note the revised affiliation in the manuscript:

Previous: Department of Science and Humanities, National University of Computer and Emerging Science, Chiniot-Faisalabad Campus, Chiniot, Pakistan

Revised: FAST School of Management, National University of Computer and Emerging Science, Chiniot-Faisalabad Campus, Chiniot, Pakistan

We appreciate your attention to this update.

Query. Review Comments to the Author

Query # 4. Have the authors made all data underlying the findings in their manuscript fully available?

Response: Reviewer #1: Yes

Reviewer #2: No

Response to Reviewer #2 – Query #4:

Thank you for highlighting the PLOS ONE Data Availability Policy requirements. We fully acknowledge the journal’s policy regarding open and unrestricted access to the data underlying the findings of the manuscript. In compliance with this policy, we have ensured that all relevant underlying data supporting the results of this study are now provided as a Supplementary File.

Reviewer #1: Minor comments

Query: The sentences in Lines 87-90 should be cited as well as sentences in line 99-102. Line 348 states 'Future longitudinal and cross-cultural studies may further clarify these relationships'. This statement is ambiguous and so there should be a mini section that addresses the future research directions in the area of this research.

Response: We have added appropriate citations for the sentences in Lines 87-90 and Lines 99-102. Additionally, we have revised the statement in Line 295 on future research directions in the manuscript following page # 16; line # 295-297.

Reviewer #2: Thank you for addressing the issues and doing modifications accordingly.

My major concern was the generalizability of the results of a tertiary hospital data, where cases can differ from global estimates and more aggressive disease form are present. Since the authors declared that the data do not provide generalizable data at the moment and more studies are needed, then the main issue has been addressed and paper is ready for publication.

---

## [Editor Report · Decision Letter 2]

16 Mar 2026

Association of Marital and Parental Status with Stress, Support, Adherence, and Quality of Life Among Breast Cancer Women

PONE-D-25-55850R2

Dear Dr. Shah,

We’re pleased to inform you that your manuscript has been judged scientifically suitable for publication and will be formally accepted for publication once it meets all outstanding technical requirements.

Kind regards,

Mukhtiar Baig, Ph.D.

Academic Editor

PLOS One
---

## [Editor Report · Acceptance letter]

PONE-D-25-55850R2

PLOS One

Dear Dr. Shah,

I'm pleased to inform you that your manuscript has been deemed suitable for publication in PLOS One. Congratulations! Your manuscript is now being handed over to our production team.

Kind regards,

on behalf of

Professor Mukhtiar Baig

Academic Editor

PLOS One